# The Role of ZEB2 in Human CD8 T Lymphocytes: Clinical and Cellular Immune Profiling in Mowat–Wilson Syndrome

**DOI:** 10.3390/ijms22105324

**Published:** 2021-05-18

**Authors:** Katie Frith, C. Mee Ling Munier, Lucy Hastings, David Mowat, Meredith Wilson, Nabila Seddiki, Rebecca Macintosh, Anthony D. Kelleher, Paul Gray, John James Zaunders

**Affiliations:** 1Sydney Children’s Hospital, Randwick, NSW 2031, Australia; lucyahastings@gmail.com (L.H.); david.mowat@health.nsw.gov.au (D.M.); rebecca.macintosh@health.nsw.gov.au (R.M.); Paul.Gray1@health.nsw.gov.au (P.G.); 2School of Women’s and Children’s Health, UNSW Sydney, Sydney, NSW 2052, Australia; 3The Kirby Institute for Infection and Immunity in Society, UNSW Sydney, Sydney, NSW 2052, Australia; akelleher@kirby.unsw.edu.au; 4Department of Clinical Genetics, Children’s Hospital at Westmead, Sydney, NSW 2145, Australia; meredith.wilson@health.nsw.gov.au; 5INSERM U955 Eq16, Vaccine Research Institute and Université Paris Est Créteil, F-94010 Créteil, France; nabila.seddiki@inserm.fr; 6Centre for Applied Medical Research, St Vincent’s Hospital, Darlinghurst, NSW 2010, Australia

**Keywords:** zinc finger E-box binding homeobox 2 gene *(ZEB2*), CD8 T cells, Mowat–Wilson syndrome

## Abstract

The *Zeb2* gene encodes a transcription factor (ZEB2) that acts as an important immune mediator in mice, where it is expressed in early-activated effector CD8 T cells, and limits effector differentiation. *Zeb2* homozygous knockout mice have deficits in CD8 T cells and NK cells. Mowat–Wilson syndrome (MWS) is a rare genetic disease resulting from heterozygous mutations in *ZEB2* causing disease by haploinsufficiency. Whether ZEB2 exhibits similar expression patterns in human CD8 T cells is unknown, and MWS patients have not been comprehensively studied to identify changes in CD8 lymphocytes and NK cells, or manifestations of immunodeficiency. By using transcriptomic assessment, we demonstrated that ZEB2 is expressed in early-activated effector CD8 T cells of healthy human volunteers following vaccinia inoculation and found evidence of a role for TGFß-1/SMAD signaling in these cells. A broad immunological assessment of six genetically diagnosed MWS patients identified two patients with a history of recurrent sinopulmonary infections, one of whom had recurrent oral candidiasis, one with lymphopenia, two with thrombocytopenia and three with detectable anti-nuclear antibodies. Immunoglobulin levels, including functional antibody responses to protein and polysaccharide vaccination, were normal. The MWS patients had a significantly lower CD8 T cell subset as % of lymphocytes, compared to healthy controls (median 16.4% vs. 25%, *p* = 0.0048), and resulting increased CD4:CD8 ratio (2.6 vs. 1.8; *p* = 0.038). CD8 T cells responded normally to mitogen stimulation in vitro and memory CD8 T cells exhibited normal proportions of subsets with important tissue-specific homing markers and cytotoxic effector molecules. There was a trend towards a decrease in the CD8 T effector memory subset (3.3% vs. 5.9%; *p* = 0.19). NK cell subsets were normal. This is the first evidence that *ZEB2* is expressed in early-activated human effector CD8 T cells, and that haploinsufficiency of *ZEB2* in MWS patients had a slight effect on immune function, skewing T cells away from CD8 differentiation. To date there is insufficient evidence to support an immunodeficiency occurring in MWS patients.

## 1. Introduction

The zinc finger E-box binding homeobox 2 gene (*ZEB2*; OMIM #605802) encodes ZEB2, one of a group of transcription factors important in epithelial-to-mesenchymal transition (EMT). Heterozygous mutations in *ZEB2* underlie Mowat–Wilson syndrome (MWS; OMIM #235730), a rare genetic disorder first described in 1998, with more than 300 patients reported to date [1,2,3,4]. MWS is characterized by intellectual disability and distinctive facial features, with variable multiple congenital anomalies including, microcephaly, Hirschsprung disease, epilepsy, genitourinary and cardiac defects, agenesis of the corpus callosum and short stature [2,4]. Multiple deletions of, or pathogenic variants in *ZEB2* have been identified in MWS patients, predominantly deletions and truncating mutations, which cause disease by haploinsufficiency [2,4].

Murine studies have demonstrated a critical role for ZEB2 in immune function including lymphocyte progenitor and terminal cell differentiation. In mice, Zeb2 is upregulated in activated CD8 T cells and represses expression of genes that promote differentiation to CD8 effector cells [5,6,7]. There are limited human studies on the immune effects of ZEB2, while the impact of ZEB2 haploinsufficiency on the immune system in MWS is debated. Infrequently MWS patients have splenic hypo/aplasia presenting with severe infections, however Omulisik et al. did not find differences in peripheral blood lymphocyte subsets in five patients with MWS compared with healthy controls (see Table 1) [2,5,8,9].

We studied early-activated antiviral effector CD8 T cells 2 weeks after vaccinia inoculation of healthy adults, to assess human *ZEB2* upregulation, as is seen in murine models of experimental infection. We also sought to further define the immune phenotype of patients with MWS in a cohort of six genetically confirmed patients who underwent comprehensive immune functional testing. 

## 2. Results

Six patients with genetically confirmed MWS were included in our study (five male; median age 16 y, range 3–22 years (see Table 2, and Appendix B and Appendix A for extended clinical information). All patients had a normal spleen on ultrasound. All patients had experienced at least one episode of acute otitis media (AOM) infection. Patient 1 and patient 2 experienced frequent AOM with perforation and chronic aural discharge with grommets in situ. The same two patients had significant infection histories including recurrent Hirshsprung associated enterocolitis (HAEC) with systemic sepsis and frequent episodes of bacterial pneumonia necessitating hospital admission. Patient 2 also experienced frequent bacterial sinusitis, and both had chronic thrombocytopenia. Patients 3–6 had no other significant infections. Besides chronic idiopathic thrombocytopenia in patients 1 and 2, there was no clinical or biochemical evidence to suggest autoimmune conditions in our cohort (see Appendix B and Appendix A for extended clinical information).

### 2.1. ZEB2 Expression in Early Antiviral Human Effector CD8 T Cells

We previously studied human T cell responses to vaccinia inoculation in healthy adult volunteers, which allowed us to follow the first 4 weeks of the immune response to this vaccine and identify transient activated CD38++ effector CD4 and CD8 T cells at 2 weeks following inoculation [11] that had not been reported before. We purified these transient effectors for microarray analysis [12]. Murine models have described a role for Zeb2 during the very early stages of the CD8 T cell response to experimental infections, but a similar role in human CD8 differentiation is not known [5,6]. Furthermore, changes in the ratio of ZEB2 expression to ZEB1 expression are associated with TGFß-1 signalling and differentiation during embryogenesis, wound healing and malignant progression [13,14]; reciprocal expression of ZEB2 and ZEB1 have recently been implicated in CD8 effector T cells in murine studies [15].

Figure 1A shows that the transient early-activated human CD8 effector cells (CD38++CD8) expressed ZEB2 at a greater level than they expressed ZEB1 (Figure 1B). These changes resulted in an increase in the ratio of ZEB2:ZEB1 expression above 1, compared to the ratio of less than 1 in the other subsets studied (Figure 1C). Elevated levels of ZEB2 mRNA were confirmed by quantitative real-time RT-PCR analysis of purified early-activated human CD38++ CD8 effector T cells, 2-weeks post-inoculation of vaccinia virus from two different donors (Figure 1D). The microarray data also show that the transient early-activated CD8 effector cells had higher combined expression of the TGF-ß1 receptors TGFBR1 and TGFBR3 (Figure 1E), as well as a strikingly increased expression of IGFBP7 (Figure 1E), PDGFD, KLF10, and CRIM1 and SMAD5 (Appendix A), which have all been described as associated with TGF-ß1 signalling in other cell types [16,17,18,19,20]. 

These findings are consistent with a role for TGF-ß1 signalling in the early-activated CD8 effector T cells. Additionally, the ZEB2:ZEB1 expression alterations in the transient early-activated antiviral CD8 effector cells are similar to the EMT response in some tumours [13,14].

### 2.2. Lymphocyte Subsets

Immunophenotyping of peripheral blood found no statistical differences in proportions or cell counts of lymphocytes (Figure 2) specifically T cells, B cells or NK cells (Figure 2A) between MWS patients and controls, except for patient 1, who had chronic lymphopenia. However, the CD8 T cell subset, as % of lymphocytes, was reduced in MWS patients compared to controls studied on the same day (median 16.4% vs. 23.3%, respectively; *p* = 0.028; Figure 2A). The proportion of NK cells as % of lymphocytes in MWS patients was slightly increased (18% vs. 11.3%; *p* = 0.045; Figure 2A). There was a trend to a slight decrease in CD8 T cell counts/µL in MWS patients compared to a large number of historical controls (median 459/µL vs. 596/µL, respectively; *p* = 0.13; Figure 2B). 

Detailed analysis of CD8 T cell subsets (Appendix A) defined by functional, trafficking and differentiation markers found that MWS patients had a slight, but not significant, reduction of the T effector memory subset (CD45RO+CD62L−) as % of CD8 T cells compared to contemporaneous controls (16.0% vs. 26.3% and *p* = 0.132; Figure 3A). When expressed as % of lymphocytes, a non-significant trend was found for this subset in MWS compared to either contemporaneous or historical controls (median 3.3% vs. 6.6% vs. 5.8%, respectively; Figure 3B). Both CD127+ and CD127− CD8 Tem cells were slightly lower in MWS patients (Figure 3B). 

Differentiation of CD8+ effector T cells was assessed based on expression of the cytotoxic effector molecules Granzyme B and Perforin, and was comparable between MWS patients and healthy controls (Figure 3C). Similarly, there were no clear differences in terminally differentiated CD45RA+CD62L−CD28−CX3CR1+ cells, or in MAIT cells (CD161^high^CCR6+), or in gut-homing (CD45RO+CD49d+Integrinß7+) CD8 subsets between MWS patients and controls (Figure 3D).

### 2.3. Lymphocyte Function

We tested the ability of CD8 T cells to respond in whole blood cultures to the polyclonal mitogen SEB, using our OX40 flow-based assay as the readout. The results show that CD8 T cells in samples from MWS subjects responded to SEB at levels that were not significantly different compared to controls, as shown in Figure 4.

We also tested CD4 recall responses to standard CD4 antigens, including CMV and tetanus toxoid, and CD4 T cells in samples from MWS subjects responded to these antigens at similar levels to controls (not shown).

## 3. Discussion

The *ZEB2* gene encodes a multi-functional transcription factor known to play a significant role in early neurogenesis and EMT [21]. More recently ZEB2 was demonstrated to have a role in hematopoietic stem cell differentiation, and in mice, ZEB2 is the most highly induced transcription factor during NK cell maturation, with the frequency of mature NK cells being proportional to ZEB2 expression. ZEB2 is also essential for the differentiation and survival of mature murine NK cells [22,23,24]. ZEB2 is upregulated in activated murine CD8 T cells and is an important part of a broader transcriptional network responsible for establishing a balance in the terminal differentiation of CD8 cells between memory and effector CD8 T cells, by repressing memory gene expression [5,15,25]. 

We report for the first time that ZEB2 is expressed in human early-activated effector CD8 T cells. This was possible through our studies of the very early response to vaccinia virus inoculation [11,12]. Furthermore, microarray analysis also revealed relative concomitant down-regulation of ZEB1 in the same cells, consistent with the well described TGF-ß1 induced EMT associated with neoplastic transformation [14]. However, we did not observe upregulation of expression of other mediators of EMT, including SNAIL1, SNAIL2 or TWIST1 [14]. 

Heterozygous mutations or deletions involving *ZEB2* cause Mowat–Wilson syndrome (MWS), with more than 300 patients reported [2]. The majority have either gene deletions or mutations resulting in absent or truncated/non-functioning protein production leading to ZEB2 haploinsufficiency [2]. There are scant published data on the effect of ZEB2 haploinsufficiency on adaptive immune function in MWS patients (Table 1). Omulisik et al. studied the peripheral blood lymphocytes of five MWS patients as representatives of cellular immune function in ZEB2 haploinsufficiency but no clinical history was provided [5]. In contrast to their findings in mice, the MWS patients had no significant difference in the frequencies of CD4, CD8, naïve central memory and effector memory CD8 T cells compared with controls, and observed a trend towards a reduced percentage of CD19 B cells only [5]. In contrast, expansion of terminally differentiated CD8 cells (CD8^+^CCR7^−^CD45RA^+^) associated with reduced memory B cells, switched memory B cells and CD4 memory cells was observed in the only report of hypogammaglobulinaemia in MWS [9]. This patient had profound pan-hypogammaglobulinaemia, a normal spleen and no history of significant infections. Given these reports we sought to better define the involvement of ZEB2 in human adaptive immune function, and the effect of ZEB2 haploinsufficiency on infectious predisposition and adaptive immune phenotype in six patients with genetically defined MWS. 

In our MWS cohort two patients had significant infection histories (patients 1 and 2). Both demonstrated recurrent, severe sinopulomonary infections in association with chronic thrombocytopenia. Whether their infection predisposition is related to defective adaptive immunity or structural factors predisposing to infections is debatable. AOM is reported in approximately 35% MWS patients, including all patients in our cohort, but we did not find altered expression of the main adhesion and trafficking markers, such as mucosal trafficking integrin alpha4+beta7+ CD8 T cells [2]. Additionally, there were normal proportions of terminally differentiated granzyme B+ perforin+ cytotoxic CD8 T cells. It is possible that structural factors, such as facial abnormalities and HD, both common in MWS, predispose patients to specific infections [2,26]. While all our patients reported episodic AOM, patients 1 and 2 had recurrent AOM with perforation, and patient 1 has a submucosal cleft palate. HD associated enterocolitis (HAEC) with sepsis is well recognised, occurring in up to 60% of HD patients, and two of our three patients with HD suffered this complication [27]. 

While immune deficiency and dysregulation have not been typically associated with MWS patients, hypo/asplenia has been reported in six MWS patients presenting with invasive pneumococcal infections [2,8,9,26]. All cases of hypo/asplenia in MWS patients have been reported in females, despite MWS having a slightly higher male prevalence, however the numbers are too small to determine if there is a gender bias towards asplenia in MWS. That said, asplenia is a rare occurrence in MWS, reported in only 1 of 76 patients in the largest published MWS cohort, and not found in any of our patients, or reported in other MWS cohorts [2,3].

Although none of our patients had asplenia or documented invasive pneumococcal disease, patients 1 and 2 had recurrent severe pneumonias, as well as recurrent AOM with perforation with *Pneumococcus* species cultured. Our patients all had normal IgG levels with generally poor baseline titres to specific pneumococcal serotypes, despite all receiving complete primary immunisation with three doses of pneumococcal conjugate vaccine (13vPCV or 7vPCV depending on their age, Appendix A). The two patients with significant infection histories had generally poor baseline responses to all 14 pneumococcal serotypes tested, although one had a high baseline titre to serotype 10A indicating prior infection. All patients were offered Pneumovax 23 to assess for specific antibody deficiency, but consent was only obtained for patients 1 and 6. Both demonstrated excellent post vaccination responses to non-conjugated pneumococcal vaccination, including patient 1 with severe recurrent sinopulmonary infections, indicating normal polysaccharide antigen responses Appendix A. 

None of our cohort had significant viral infections or haemophagocytic lymphohistiocytosis, which might be seen in the context of a CD8 T cell or NK cell defect. Although ZEB2 has an important role in murine NK cell maturation we found a small but significant increase in mature NK cell numbers in our ZEB2 haploinsufficient patients. In addition, lymphocyte subset analysis demonstrated haploinsufficiency of ZEB2 was associated with slight but significant skewing of T cell subsets away from CD8 T cells in our MWS cohort. Nevertheless, we did not find any major abnormalities in an exhaustive range of important subsets of memory CD8 T cells, defined by differentiation, trafficking and functional markers. In vitro responses to the polyclonal mitogen SEB were also normal, suggesting there was no broad CD8 dysfunction. 

We observed normal CD19 numbers in three of our MWS cohort, but mildly elevated in one (patient 5: 0.95 × 10^9^/L; reference range: 0.2–0.6 × 10^9^/L) and reduced in two (patients 1: 0.05 × 10^9^/L and 2: 0.12 × 10^9^/L; reference range: 0.16–0.36 × 10^9^/L). Both patients with reduced CD19 numbers had significant infection histories, and chronic thrombocytopenia. Autoimmunity, in particular, autoimmune cytopenias occur more frequently in primary immunodeficiencies, especially T cell deficiencies [28]. Confirming an autoimmune aetiology for thrombocytopenia was beyond the scope of our study and other factors, such as medications, need to be considered. Thrombocytopenia is a recognised side effect of sodium valproate and carbamazepine which patients 1 and 2 were treated with, respectively. No other patients had evidence of clinically significant autoimmunity or inflammation. Patient 4 had an elevated ANA (1:1280) with positive dense fine speckle 70 antibodies (DFS70) antibodies, with no clinical features of autoimmune disease. DFS70 antibodies are found in up to 22% of healthy people and have a negative association with ANA associated rheumatic disease [29]. Patient 6 had an elevated gliadin IgG (160 U/mL, normal range 0–7 U/mL) with a normal gliadin IgA and tTG with no clinical manifestations of gastrointestinal disease and this was considered clinically irrelevant.

From our studies of the dynamics of human T cells following vaccinia inoculation of healthy adults [11,12], we were uniquely able to investigate the role of ZEB2 in early-activated CD8 T-cells. Consistent with the murine data, there was an upregulation of ZEB2 expression in the early-activated CD8 T-cells in volunteers at day 13 post-inoculation, as well as a concomitant decrease in ZEB1 expression. Additionally, consistent with TGFß-1 signalling in early-activated effector CD8 T cells, we found that all three TGF-ß1 receptors were highly expressed, particularly TGFBR1 and TGFBR3, which were selectively transcriptionally upregulated, as well as an increased expression of SMAD5. Furthermore, there was a striking elevation of expression of insulin-like growth factor binding protein 7 (IGFBP7), which has been reported as a direct target of TGFß-1 signalling in human renal proximal tubular epithelial cells [20], but, to our knowledge, has not previously been reported in human T lymphocytes. Similarly, KLF10, previously named TGF-β inducible early gene 1 (TIEG1) [30], was expressed at a much higher level in the early-activated effector CD8 T cells. Expression of KLF11 (TIEG2) and KLF4 and KLF8 were also slightly higher (Appendix A) compared with other T cell subsets. Additionally, PDGFD expression was very highly upregulated in the early-activated effector CD8 T cells, and PDGF family members are very closely associated with TGFß1 in fibrotic disease [17].

Overall, our results provide further supporting evidence that the TGF-ß1 signalling/SMAD pathway is potentially important in human antiviral CD8 T cell responses in vivo. One previous study suggested that TGFß-1 enhanced human CD4 CTL activity in vitro via suppression of Eomes transcription factor [31], but does not provide evidence of TGFß-1 affecting CD8 T cell differentiation in vivo. The microarray data for elevation of ZEB2 in early-activated CD8 T cells following vaccinia virus inoculation were confirmed by quantitative real-time PCR, but not so far at the protein level due to lack of suitably specific antibodies. Further studies on early-activated human CD8 T cells are planned, particularly given the urgent need to identify the role of these cells in SARS-CoV-2 infection. 

Murine models have shown that TGF-ß1 signalling in early short-term effector CD8 T cells is associated with their apoptosis during the contraction phase of an immune response, and counterbalances effector cell survival by IL-15 signalling [32]. We previously showed that early-activated human effector CD8 T cells dramatically downregulate the IL-7 receptor alpha chain and anti-apoptotic Bcl-2, and this may also be due to TGFß1 signalling [11,33,34,35,36]. However, while upregulation of ZEB2 definitely appears to be involved in human anti-viral CD8 T cell differentiation, its effect is not enough to result in significant immunological abnormalities in patients where ZEB2 haploinsufficiency was enough to induce developmental abnormalities in other organs. In addition, no significant humoral immune abnormalities or autoimmunity were found in our cohort. The difference between murine knockout models and human patients remains to be determined.

## 4. Materials and Methods 

### 4.1. Subjects

Patients with MWS were recruited from our institution and consent was obtained from their parents. Ethics approval to collect data and perform functional immune assessments (blood samples and abdominal ultrasounds) for these patients was granted by the Sydney Children’s Hospitals Network Human Research Ethics Committee (HREC) LNR/15/SCHN/195.

All MWS patients underwent: abdominal ultrasound to assess splenic size and peripheral blood samples for immune testing; a parental questionnaire regarding the child’s history of infections was completed. Immune testing results were compared with healthy age-matched controls and earlier results if available. As baseline serotype specific antibody responses were generally poor, all six patients were offered Pneumovax (23vPPV) with repeat serotype specific antibody testing to further characterise this deficiency, however parental consent was only obtained on behalf of two patients. Thyroid antibodies tested included thyroglobulin (anti-TG) and thyroid peroxidase antibodies (anti-TPO). Celiac antibodies tested were gliadin IgG and IgA and tissue transglutaminase IgA (tTG). Immunoglobulin level assessment was performed as a standard clinical test through the NSW Health Pathology-Randwick laboratory. 

Healthy adults underwent primary vaccinia virus immunisation (VV; Dryvax vaccine, Wyeth Laboratories, Marietta, PA, USA) as previously described [11,12], and had ACD and sodium heparin (NaHep) anti-coagulated peripheral blood drawn at day 13 post-inoculation for PBMC separation. The VV studies were approved by the Australian Red Cross Blood Service HREC, St Vincent’s Hospital Sydney HREC (HREC/10/SVH/130) and the University of Melbourne Health Sciences Human Ethics Sub-Committee (Ethics ID:1035129). Healthy adult controls for detailed T cell subset immunophenotyping were recruited from hospital and university staff (St Vincent’s Hospital HREC ID: HREC/13/SVH/45) [37]. All subjects gave informed written consent. 

### 4.2. Flow Cytometry

An amount of 100 µL of fresh Na heparin anti-coagulated peripheral blood samples (<4 h after venesection) were stained with monoclonal antibodies (mAb) according to the manufacturers’ directions, incubated for 15 min at RT, lysed with Optilyse C (Beckman Coulter, Brea, CA, USA) for 10 min at RT, washed with 2 mL PBS and resuspended in 250 µL 0.5% paraformaldehyde in PBS, as previously described [12,37]. Intracellular staining for cytotoxic granules and granzymes was performed using FACSLyse (BD Biosciences, San Jose, CA, USA) and FACSPerm 2 (BD Biosciences) as previously described [12]. The mAb’s used in this study are listed in Appendix A. Fixed samples were kept at 4 °C before 14-colour analysis, within 24 h, on an LSR II (BD Biosciences) as previously described [12,37,38].

### 4.3. Lymphocyte Function

CD8+ T lymphocyte function was measured in vitro as upregulation of CD25 and CD134 (OX40) after 44–48 h whole blood cultures (OX40 assay) as previously described [39]. Briefly, 0.25 mL fresh Na heparin blood was added to 0.25 mL Iscove’s modified Dulbecco’s medium in 24-well plates. Individual cultures were incubated with either medium only (negative control) or with staphylococcal enterotoxin B (SEB; 1 µg/mL; Sigma Aldrich, St Louis, MO, USA). At the end of the culture, 100 µL of each culture was stained with CD3-PerCP-Cy5.5, CD4-AlexaFluor700, CD25-APC and CD134-PE (all from BD Biosciences) and analysed on an LSRII flow cytometer (BD) as previously described [39].

### 4.4. Cell Sorting, RNA Extraction, Preparation of cRNA for Microarray Analysis

Cell sorting of CD3+CD8+ CD45RAnegCD38++ activated effector T cells from PBMC from two vaccinated adults at day 13 post-inoculation (VV1 and VV2), for microarray analysis has been previously described [12], data are accessible at NCBI GEO database accession GSE161037. Sorted cells were lysed in TRIzol^®^ (Invitrogen, Carlsbad, CA, USA) then stored at −80 °C until RNA extraction was performed using a small-scale RNA extraction method as previously described [40]. Briefly, cDNA was specifically transcribed from 500 ng of mRNA using a poly-T nucleotide primer, containing a T7 RNA polymerase promoter (GeneWorks, Adelaide, Australia). Biotinylated, antisense target cRNA was subsequently synthesized by in vitro transcription, using the Two-Cycle Target Labelling kit (Affymetrix, Inc., Santa Clara, CA, USA). Fifteen micrograms of biotin-labelled target cRNA was then fragmented, and used to prepare a hybridisation mixture, which included probe array controls and blocking agents. Hybridisation to U133A Plus 2.0 (HG-U133 Plus 2.0; Affymetrix, Santa Clara, CA, USA) arrays was conducted for 16 h at 45 °C and 60 rpm. After hybridisation, washing and staining of the hybridised probe array was performed by an automated fluidics station, according to the manufacturer’s protocols. The stained probe array was scanned using the Agilent (Palo Alto, CA, USA) GeneArray Laser Scanner, and the resultant image was captured as a data image file, which was then analysed using Microarray Analysis Suite software version 5.0 (MAS 5.0; Affymetrix, Santa Clara, CA, USA). Signal value represents the level of expression of a transcript, as previously described [41,42]. Scaling of gene expression intensity levels was performed using Microarray Analysis Suite Software 5.0 (Affymetrix, Santa Clara, CA, USA).

### 4.5. Real-Time PCR

CD3+CD8+ CD45RO+ CD38++ activated effector T cells were cell sorted from cryopreserved PBMC from two vaccinated adults at day 13 post-inoculation (VV3 and VV4), as well as CD38neg CD45RO+ resting memory CD8+ T cells and CD38neg CD45ROneg CD8 T cells, using a FACSAria Fusion (BD Biosciences, San Jose CA, USA). RNA was extracted using the Maxwell RSC automated extraction platform (Promega, Madison, WI, USA), with the Maxwell RSC Simply RNA Tissue kit (Promega, Madison, WI, USA), according to the manufacturer’s protocol. Copies of ZEB2 were determined using the ZEB2 PrimePCR Probe assay, based on a standard curve using the ZEB2 PrimePCR Template (BioRad, Hercules, CA, USA), and using One-step PrimeScript RT-PCR Reagent Kit (Takara Bio, Shiga, Japan), all according to the manufacturers’ protocols. 

### 4.6. Statistical Analysis

Results from MWS patient group and from controls, respectively, are expressed as medians and interquartile ranges, and were compared by Mann–Whitney unpaired non-parametric test using PRISM 7 (GraphPad Software, San Diego, CA, USA). *p*-values < 0.05 were considered statistically significant. 

## 5. Conclusions

Based on our cohort and published reports, immune deficiency appears to be an infrequent association with MWS. We demonstrated ZEB2 expression in early-activated CD8 T cells, and a corresponding reduction in the proportion, but not function, of CD8 cells in ZEB2 haploinsufficiency. None of our cohort had severe viral infections or HLH. Humoral immunodeficiency is rarely reported in MWS, and was not found in our cohort. Overall our findings suggest that infections observed in MWS patients, including frequent AOM, and more rarely HAEC and pneumonia, are more likely due to structural anomalies and physical factors than impaired immune function. Hypo/asplenia is a rare developmental anomaly in MWS and there does not appear to be a genotype:phenotype correlation. However, given the associated risk of severe infection and options for immunisation and prophylaxis we would agree with previous recommendations to screen for splenic aplasia with ultrasound in all newly diagnosed MWS patients [2]. We would also recommend baseline immunoglobulin and lymphocyte subset measurement at diagnosis, with repeat measurement if clinically indicated, given the rare but serious reports of immunodeficiency in MWS and the simplicity of this test. We would not advise routine screening for autoimmunity based on findings in our cohort and published reports. 

## Figures and Tables

**Figure 1 ijms-22-05324-f001:**
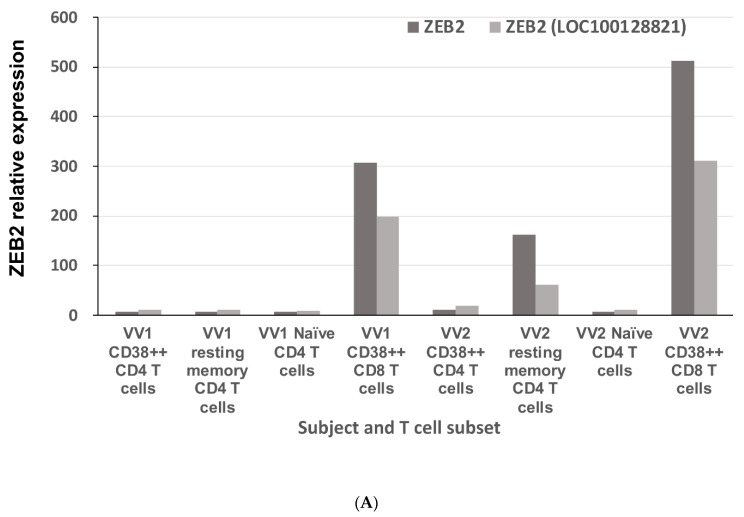
Gene expression profiling of ZEB2 and ZEB1 RNA and TGF-ß1 signalling molecules in microarrays of early-activated effector CD38+++ CD8 T cells following vaccinia inoculation. Four purified T cell subsets were cell sorted from fresh PBMC: CD38++ CD4 T cells; resting memory CD4 T cells; naïve CD4 T cells and CD38++ CD8 T cells from each of two healthy adult donors, VV1 and VV2. RNA levels for each subset were normalised and calculated in Affymetrix MAS 5.0 software. (**A**) Elevated expression of ZEB2 in microarrays of early-activated human CD38++ CD8 effector T cells, 2-weeks post-inoculation of vaccinia virus. The light grey bars are for probe set 235593_at, annotated as specific for Zeb2 (Accession No. AL546529) and the dark grey bars are for probe set 203603_s_at (Accession No. NM_014795), also annotated as Zeb2. (**B**) Reduced expression of ZEB1 in microarrays of early-activated human CD38++ CD8 effector T cells, 2-weeks post-inoculation of vaccinia virus. The light grey bars are for probe set 212758_s_at, annotated as specific for Zeb1 (Accession No. AI373166) and the dark grey bars are for probe set 212764_at (Accession No. AI806174), also annotated as Zeb1. (**C**) Elevated ZEB2 and lower ZEB1 expression in early-activated human CD38++ CD8 antiviral effector cells leads to a ratio of >1 in those cells, compared to other T cells. (**D**) Elevated levels of ZEB2 mRNA were confirmed by quantitative real-time RT-PCR of purified early-activated human CD38++ CD8 effector T cells, from cryopreserved PBMC from two different donors, 2-weeks post-inoculation of vaccinia virus. (**E**) Elevated expression of TGF-ß1 receptors (TGFBR1 and TGFBR3) and IGFBP7 in early-activated human CD38++ CD8 antiviral effector cells, by microarray analysis.

**Figure 2 ijms-22-05324-f002:**
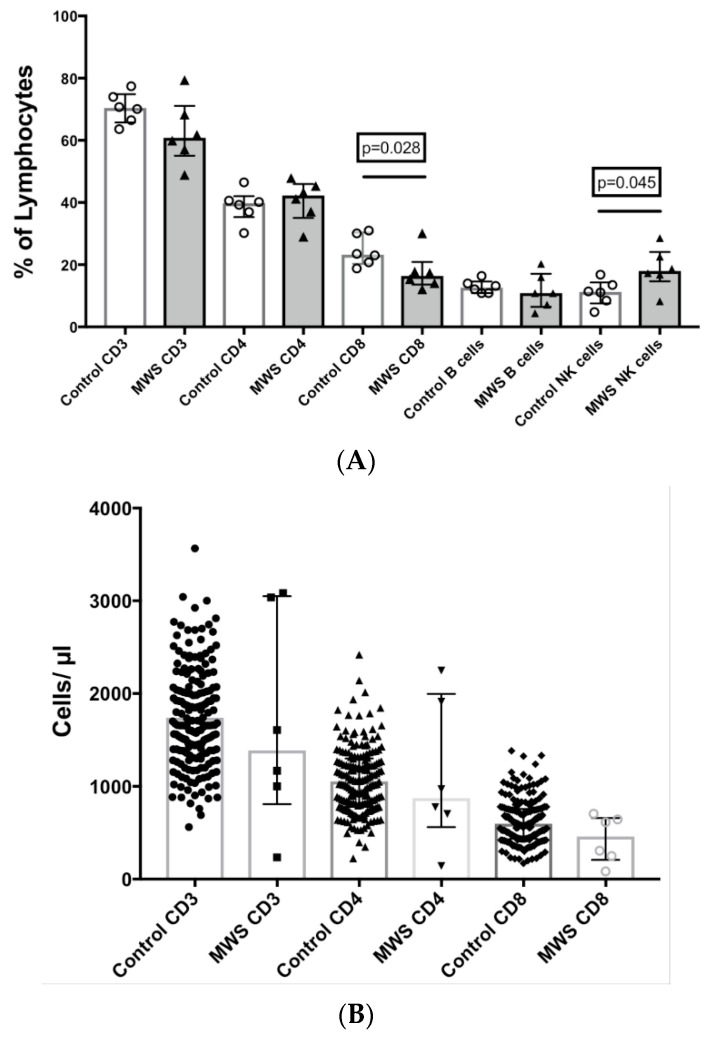
Comparison of lymphocyte subsets in peripheral blood between controls and patients with Mowat–Wilson syndrome. (**A**) T cell (CD3+, CD4+ and CD8+), B cell (CD19+) and NK cell (CD56+) subsets as % of total lymphocytes. (**B**) Comparison of T cell counts (cells/μL) between controls and patients with Mowat–Wilson syndrome.

**Figure 3 ijms-22-05324-f003:**
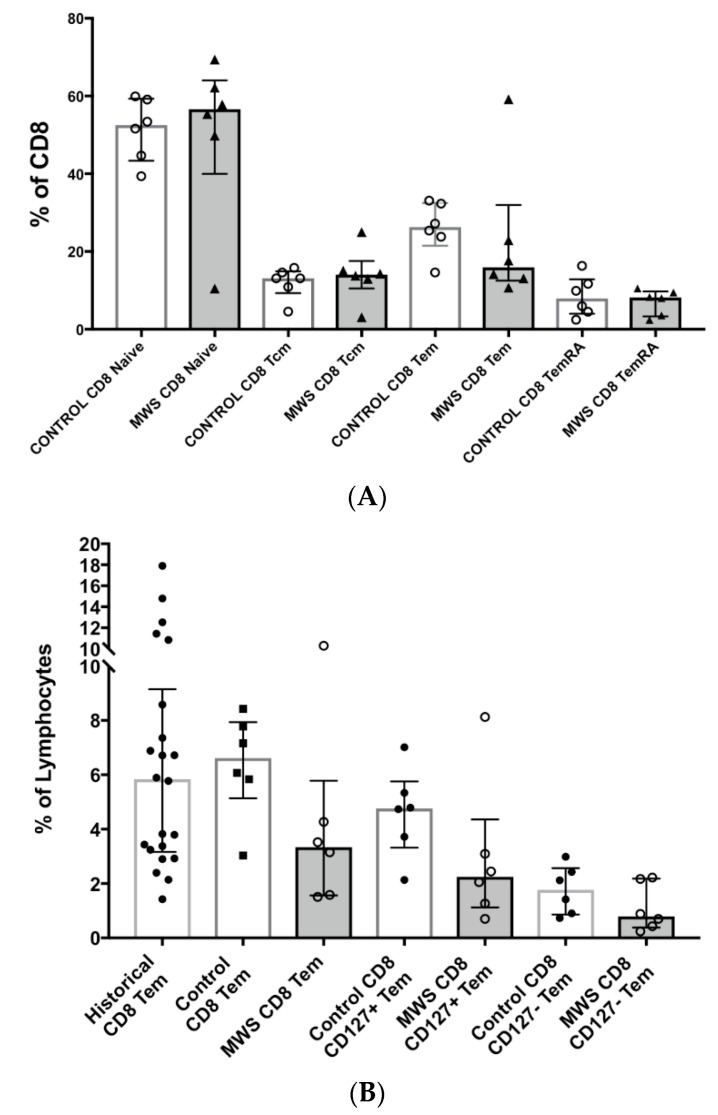
Comparison of CD8 T cell subsets in peripheral blood between controls and patients with Mowat–Wilson syndrome. (**A**) Comparison of naïve (CD45RO−CD62L+), central memory (CD45RO+CD62L+), effector memory (CD45RO+CD62L−) and TEMRA (CD45RO−CD62L−) subsets of CD8 T cells between controls and patients with Mowat–Wilson syndrome. Subsets are expressed as % of total CD8 T cells. (**B**) Comparison of CD8 T effector memory (CD45RO+CD62L−) subsets between historical controls, same day controls and patients with Mowat–Wilson syndrome. Subsets are expressed as % of total lymphocytes. (**C**) Comparison of cytotoxic CD8 T effector cells as % of naïve (CD45RO−CD62L+), central memory (CD45RO+CD62L+), effector memory (CD45RO+CD62L−) and TEMRA (CD45RO−CD62L−) subsets, respectively, controls and patients with Mowat–Wilson syndrome. Granzyme B+ Perforin+ cells are expressed as % of each subset of CD8+ T lymphocytes. (**D**) Comparison of terminally differentiated (CD45RA+CD62L−CD28-CX3CR1+) CD8 T cells, MAIT (CD45RO+CD161highCCR6+CD127high) cells and gut-homing (CD45RO+CD49d+Integrinß7+) CD8 T cells between controls and patients with Mowat–Wilson syndrome. Subsets are expressed as % of total CD8 T cells.

**Figure 4 ijms-22-05324-f004:**
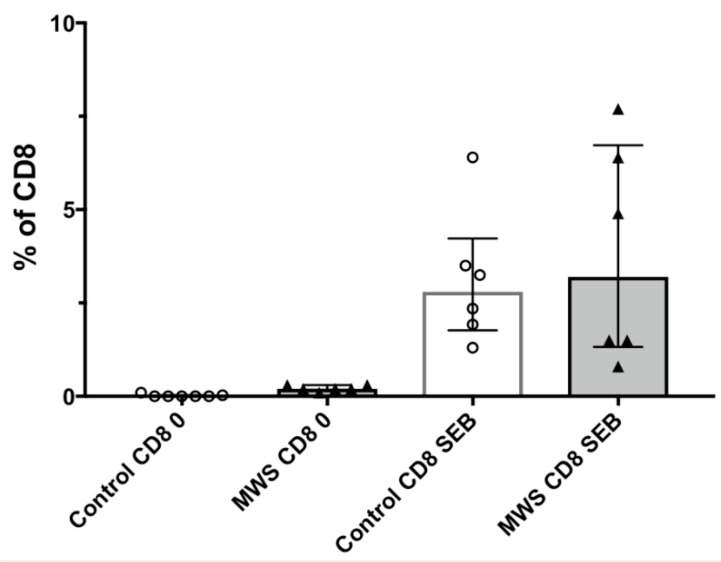
Lymphocyte function of CD8 T cells in response to the polyclonal mitogen SEB. Cells were incubated for 2 days with or without SEB, and the % of CD8 T cells that responded by upregulated coexpression of CD25 and CD134.

**Table 1 ijms-22-05324-t001:** Published reports of immune deficiency in MWS.

	Age (y)/Gender (M/F)	*ZEB2* Mutation	Infections	Spleen	Immune Function Tests	
				Hypo/asplenia	Humoral	Cellular
**Nevarez Flores** **2019 [9]**	0.8 F	c.1426dup	*S. pneumoniae* meningitis and purpura fulminans	Asplenia	NR	NR
**Sgruletti 2016 [10]**	37 F	4.6 Mb microdeletion 22q (22q22.3–22q23.2)	Nil significant	No	-Severe pan-hypogammaglobulinaemia; -Absent functional IgG response to tetanus	-Increased B cell numbers; -Dramatically reduced switched and non-switched memory B cells; -Reduced CD4 memory cells; -Increased terminally differentiated CD8 cells
**Pons** **2014 [8]**	0.7 F	c.2083C > T	*S. pneumoniae* purpura fulminans	Asplenia	NR	NR
	1 F	c.600_640dup	*S. pneumoniae* meningitis	Asplenia	NR	NR
	2 F	c.1762G > T	*S. pneumoniae* sepsis ×2	Hyposplenia	NR	NR
	F	c.1426dup	Nil reported	Hypoplasia	NR	NR
**Zweier** **2005** [4]/ **Ivanovski 2018 [2]**	1 F	c.696C > G	Ni reported	Asplenia	NR	NR

NR not reported.

**Table 2 ijms-22-05324-t002:** Immunological features of MWS participants.

	Age (y) Gender (M/F)	*ZEB2* Mutation	Infections			Splenic Hypo/ Asplenia	Immunoglobulins (g/L)			Lymphocyte Subsets (×10^9^/L)				Vaccine Responses *			Other Clinical History
			Sinopulmonary	Severe viral	Opportunistic	Yes/No	IgG	IgA	IgM	CD4	CD8	CD19	CD16+CD56	Protein	Polysaccharide (pre & post *Pneumovax 23*)		
1	22 M	NM_014795.4:c.1426dupA	AOM^p+^, pneumonia^+^, sepsis^+^	No	Candida (Inv, Sup^+^)	N	11.5	3.5	1.7	0.2 ↓	0.1 ↓	0.05 ↓	0.2	BR	Poor	AR	ID, HD with colitis; thrombocytopenia; epilepsy; tetralogy of Fallot; submucosal cleft palate; recurrent encephalopathy with autonomic dysfunction; osteopenia and #
2	21 M	partial deletion of chromosome 2q22	AOM^p+^, sinusitis^+^, pneumonia^+^, sepsis	No	Candida (Sup^+^)	N	11.5	3.6	0.2 ↓	1.2	0.8 ↑	0.1 ↓	0.2	BR	Poor	ND	ID, HD with colitis; thrombocytopenia; epilepsy; scoliosis; #; iron deficiency anaemia; neurogenic bladder; atopy
3	20 M	NM_014795.4: c.2798delC: p.Pro933HisfsTer44	AOM	No	No	N	11.5	2.2	0.6	1.1	0.4	0.3	0.5 ↑	AR	BR	ND	ID; epilepsy
4	12 F	c.690–721del	AOM	No	No	N	7.9	0.8	1.1	0.9	0.4	0.3	0.4	poor	poor	ND	ID; epilepsy; constipation
5	11 M	NM_014795.4: c.3211T > C: p.Ser1071Pro	pneumonia, tonsillitis^+^	No	No	N	9.7	1.6	1.6	2.7	1.9	0.95 ↑	0.8	AR	poor	ND	DD; epilepsy
6	3 M	NM_014795.4: c.1106T > G: p.Leu369Ter	AOM	No	No	N	7.4	0.3	0.6	2.6	1.6	0.8	0.8	AR	poor	AR	ID;HD; L esotropia

AOM acute otitis media; p denotes with perforation; + denotes recurrent infections; Inv invasive; Sup superficial; * Vaccine responses: baseline antibody titres to protein antigens (tetanus, diptheria and *H. influenza*) and polysaccharide antigens (both before and after Pneumovax 23 offered as a part of this study). More detailed vaccine responses are contained in Appendix A; ND not done; BR borderline responses; AR adequate responses; HD Hirschsprung disease; ID intellectual disability; # fractures; ↑ indicates result is above upper limit for age; ↓ indicates result is below lower limit for age.

## Data Availability

The microarray data of sorted CD4 and CD8 T cells following vaccination with vaccinia virus has been previously described [12], data are accessible at NCBI GEO database accession GSE161037.

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
