# Peer review of "The Role of ZEB2 in Human CD8 T Lymphocytes: Clinical and Cellular Immune Profiling in Mowat–Wilson Syndrome"

_ijms, 2021, doi:10.3390/ijms22105324_

Round 1

Reviewer 1 Report

The authors have sufficent addressed my comments with the new data they have supplied. This paper is now suitable for publication.

Author Response

The authors have sufficiently addressed my comments with the new data they have supplied. This paper is now suitable for publication.

We thank reviewer 1 for their comments and appreciate the time they have taken to review this manuscript.

Reviewer 2 Report

Based on the fact that In mice, Zeb2 is upregulated in activated CD8 T cells and represses expression of genes that promote differentiation to CD8 effector cells, the authors analyze expression of ZEB2 and ZEB1 in healthy subject recently exposed to vaccinia virus, showing that it is expressed in recently activated CD8 T cells were insulin-like growth factor binding protein 7 (IGFBP7) or KLF10/ TIEG1 (TGF-β inducible early gene 1) are also upregulated. In general the advantage of using microarray technology to analyzed such a small set of genes is not clear.

In this sense the author cannot claim “strong circumstantial evidence for the involvement of TGF-B1 in antiviral CD8T cells responses” just due to the fact that some of their targets are upregulated.

They then analyze cells from patients with Mowat Wilson syndrome (MWS) caused by a heterozygous mutations in ZEB2 and where immune abnormalities have not been consistently reported. They find no significant differences in cell subsets or in CD8 response to SEB between MWS patients and healthy donors.

For the FACS results it would be useful to see the actual histograms in addition to the bar graphs as there is more to the interpretation of FACS results than just the % and or the total cell count. There is no indication as to how cells have been counted.

Labeling mistake: The legend inside figure 1B is the same for both bars. Although it is explained in the figure legend it should be stated in the insert like in fig. 1A.

Author Response

This manuscript is a resubmission of an earlier submission. The following is a list of the peer review reports and author responses from that submission.

Round 1

Reviewer 1 Report

The authors provide an analysis of T cell subset proportion and activation after vaccination in 6 patients with Mowat Wilson Syndrome. This is an important question and it is where most interest in this manuscript will lie. Other aspects that are shown in Figure 1, i.e. Zeb1 and Zeb2 upregulation in healthy adult T cells after vaccination, and a too-limited analysis of TGFB-relevant genes, are inadequately presented and analysed, and add little to the manuscript (in fact if I’m honest they actively detract). I would support removal of Figure 1 unless the authors are willing to provide a proper analysis of the microarray data. I don’t think it is essential to include this data. The MWS patient data is much more robustly performed.

The manuscript is unclear in parts.

Table 1 includes colours, symbols and abbreviations that are not defined in the legend. The polysaccharide vaccine response columns were unclear to me until I got to the discussion. Please describe clearly in the legend that patient titers were measured before and after polysaccharide antigen vaccination offered as part of this study.

Table 2 is not labelled table 2, and it has no legend to define the abbreviations etc.

Figure 1 is quite hard to read in both presentation and content. There are disconnects between the figure and figure legend, eg +++ vs ++ for CD38 cells. The legend does not contain enough detail to understand this figure, evidenced by many of my questions.

In Figure 1A, it is unclear what the difference is between the groups (light and dark grey). Both are labelled as ZEB2. The axis is labelled as relative expression – relative to what? The figure contains undefined abbreviations (VV-1 and VV2). There is no measure of data spread (standard deviation etc). How many healthy adult samples were measured or are depicted here?

 Figure 1B states reduced expression of Zeb1? Compared to what? There are no statistics shown, is this a significant increase? No measure of data spread. Again it is unclear what the difference is between the two coloured bars.

Figure 1C – No measure of data spread. Unclear how these were generated – is it an average of pooled samples from figures above?

What is the rationale for comparing ZEB2 to ZEB1? Since the expression of ZEB1 seems to change independently of ZEB2 following vaccination, it is unclear why a ratio would be informative here given the paper’s focus on ZEB2. The relevance of ZEB1 to this paper, the patients, the phenotypes being investigated is not understandable at this point in the manuscript. Please include a little of the story as you introduce the experimetns, to allow a reader to know why you’re doing what you’re doing.

Figure 1D – elevated expression of TGF-B1 receptors is reported. Elevated compared to what? Is there any statistical testing to support this? Do the cells express TGFBR2 which is also a ligand for TGFB1?

The methods contains no detail on how the microarray data was pre-processed or analysed. This is not up to standard. It also does not contain an accession number, so I was unable to assess these data as part of the review process. The data if included must be uploaded to a public database upon acceptance, and a reviewer accession code should be provided to allow assessment of the data.

Regarding figure 3A, is the datapoint with no naïve T cells the same datapoint expressing 60% effector-memory T cells? It seems likely this patient had an active infection, and I wonder if it may be logical to exclude them from the analysis or present them separately.

A “slightly reduced”  T-EM subset is not a scientifically valid conclusion to draw based on p=-.132 which is not close to being statistically significant. In this situation, unless there are outliers you have reason to exclude (which you may have), you need to accept the null hypothesis, i.e. we accept any difference is likely due to chance alone. See also, the “non-significant trend”.

Line 258, regarding CD19 numbers. The comparison range is not reported anywhere for this data, so it is not possible at a glance to appreciate how different the abnormal figures are.

Line 276: “we found that all 3 TGF-ß1 receptors were highly expressed, particularly TGFBR1 and TGFBR3, which were selectively upregulated, as well as an increased expression of SMAD5.”  – no data is presented for TGFBR2, or SMAD5. There is no data presented to show that TGFBR are selectively upregulated. There is no data even to really show upregulation. It is just expression. The axes show relative expression but it is unclear relative to what.  

In the discussion, lines 282-285, there is a lot of new transcriptional data introduced for the first time which is then immediately used to draw conclusions about signalling pathways, without the reader getting to see it. This is not recommended.

Line 286: “Overall, our results show for the first time that the TGF-ß1 signalling/SMAD pathway is potentially important in human antiviral CD8 T cell responses in vivo.”    I have to strenuously disagree with both the idea that this has been demonstrated (in any way) in this manuscript, and that there is no literature on TGF-b1 pathway in human CD8 T cell responses against viruses (See Lewis et al. JCI 2016, figure 7 and others). There is no pathway analysis here to show significant enrichment of TGFb signalling pathways, no functional exploration, etc and most of the data is not shown. There is simply an expression level shown for 2 (out of 3) TGFBR genes, without any statistics. Please ensure this area of discussion is more grounded.

Methods: Several ethics approval numbers and committees are mentioned. Please clearly state which aspects of the study were covered by which ethics applications.

Please provide all information necessary to repeat the microarray analysis, including pre-processing, normalisation, expression threshold etc. Please include an accession number to allow the dataset to be reviewed.

Statistics are appropriate.  

Author Response

Please see the attachment - response to reviewer 1

Reviewer 2 Report

In this manuscript, Frith et al. examine the role of the transcription factor ZEB2 in human CD8+ T cell effector differentiation in a bid to complement data that has been generated in mice in the past decade. They also provide an immunological assessment of circulating lymphocyte populations in patients with Mowat-Wilson syndrome (MWS) who possess rare heterozygotic mutations in ZEB2. The data presented helps to support the theory that ZEB2 is involved in CD8+ effector T cell differentiation but the amount of truly novel data in this manuscript is very low. The only novel data is that ZEB2 is expressed on CD38hi CD8+ T cells following vaccinia inoculation. The data in MWS patients just corroborates the prior art and acts to demonstrate that ZEB2 is still functional even when copy of the gene is deleted in these patients. As a result, I feel the manuscript is only acceptable for publication if the authors can provide additional data to support the role of ZEB2 in CD8+ T cell differentiation further. My comments are as follows:

  1. Following vaccinia inoculation of healthy adult volunteers, ZEB2 is found highly expressed on CD38hi CD8+ T cells at weeks 1 and 2, in comparison to naïve, resting memory and CD38hi CD4+ T cells (Fig 1a). However, there is no data on ZEB2 expression in naïve or resting memory CD8+ T cells. Can this data be provided? These are vital controls that must be included to validate the findings. Also, ZEB2 expression does not increase on CD38hi CD4+ T cells. This is very interesting but it wasn’t investigated further. The data in this figure acts to only show the expression of ZEB2 and TGF-b1 receptors on CD38hi CD8+ T cells. It doesn’t divulge further on the role of ZEB2 in human CD8+ T cells. Thus, given that the main story revolves round this role, the manuscript needs more novel data on ZEB2 in human CD8+ T cells in order to be suitable for publication.
  2. The microarray data on TGF-b1 receptors and IGFBP7 (Fig 1d) again requires control data on naïve and resting memory CD8+ T cells but also comes out of the blue with no fundamental reason why these genes were assessed (other than TFG-b1 is associated with EMT) and presented in the data. The data just feels like an add-on with no true follow up experiments performed. Even then, the authors postulate that this data helps “show for the first time that the TGF-ß1 signalling/SMAD pathway is potentially important in human antiviral CD8 T cell responses in vivo (lines 287-288)”. This is a wild overstatement based on data that purely shows that TGF-b1 receptors are expressed on CD38hi CD8+ T cells in this setting. Surely, a closer look at TGF-beta signalling pathways in these cells is vital in order to make this statement. This should have been done, but was not.
  3. The phenotypic data on MWS patients pretty much verifies what has been shown before; that ZEB2 heterozygosity has no or little effect on CD8 T cell subsets in peripheral blood and their functional capacity. The study by Omilusik et al showed in mice that a homozygotic mutation in ZEB2 had a profound impact on CD8+ T cell effector differentiation and antigen-specific CD8+ T cell numbers whereas ZEB2 heterozygosity did not. Thus, in the clinical setting of MWS patients, deficiency in T cells can’t explain why these patients are prone to infection. As a result, it weakens the impact of a ZEB2 story.

Minor points:

  1. Table 1: OAM is used as an abbreviation incorrectly a number of times. This should be AOM.
  2. Fig 1b. In the bar chart, the legend states that both bars are ZEB1. This must be an error? If not, please clarify what each bar means
  3. Fig 3d: MAIT cells were identified as CD45RO+ CD161hi CCR6+ but the authors should have used an antibody against TCR Va7.2 and/or a MAIT tetramer to truly define these cells further. Does any data exist with these markers? If not, I would consider removing the MAIT data

Author Response

Please see the attachment - response to reviewer 2

Round 2

Reviewer 2 Report

The revised manuscript by Frith et al provides clarification on minor comments raised in my original review. Unfortunately, some of my major comments have not been addressed. Notably, there is no included data on ZEB2 mRNA expression in naive and resting memory CD8 T cells. This data, in my opinion, is vital in order to prove that ZEB2 expression rises during human T cell effector differentiation. The authors suggest that this happens in the absence of the data because it has been shown in mice. I feel the main message of the paper has not been verified correctly and this needs to be performed in order to solidify the conclusions.